# Effect of *Tremella fuciformis* and Different Hydrocolloids on the Quality Characteristics of Wheat Noodles

**DOI:** 10.3390/foods11172617

**Published:** 2022-08-29

**Authors:** Lifan Zhang, Jie Chen, Fei Xu, Rui Han, Miaomiao Quan

**Affiliations:** 1College of Food Science and Technology, Henan University of Technology, Zhengzhou 450001, China; 2Henan Province Wheat-Flour Staple Food Engineering Technology Research Centre, Zhengzhou 450001, China

**Keywords:** *Tremella fuciformis*, hydrocolloids, noodles

## Abstract

To improve the quality characteristics of noodles and enrich nutritional value, *Tremella fuciformis* (TF) powder was incorporated into noodles. *Tremella fuciformis* (TF) is an edible fungus with rich nutritional value, and TF gel has good viscosity properties. This paper explored the effect of TF on noodle quality, and compared the difference between TF and three hydrocolloids: sodium alginate (SA), guar gum (GG) and xanthan gum (XG). The results showed that TF could significantly (*p* < 0.05) increase the hardness, adhesiveness and chewiness of noodles, and showed a decreasing trend for additions greater than 3%. The addition of 3% TF enhanced storage modulus (G′), loss modulus (G″) and elasticity of dough. The addition of 3% TF also increased α-helix and β-sheet content, and degradation temperature in noodles. Meanwhile, it elevated the deeply bound water content and retarded water mobility. In addition, the content of slowly digestible starch and resistant starch in the noodles increased with the addition of 3% TF. It was found that the effect of 3% TF on the above data was not different from the effects of the three hydrocolloids (respectively, their optimal additions), and improved the quality characteristics of the noodles. The results provide guidance for the application of TF and the development of a new natural hydrocolloid and nutritionally fortified noodles.

## 1. Introduction

*Tremella fuciformis* (TF), also known as white fungus or snow fungus, is a widely distributed fungus. It has developed into one of the most popular cultivated fungi in China and other Asian countries [1]. TF has high nutritional value and good medicinal value. In recent years, with the deepening of TF research, it has been found that many functional and physiological characteristics of TF are related to the *Tremella fuciformis* polysaccharide (TFP). TFP accounts for 60–70% of TF. It is an acidic heteropolysaccharide with mannose as the main chain. It has antioxidant, anti-aging, hypoglycemic and anti-inflammatory effects [2]. Additionally, TFP solution has good rheological and gel properties [3]. As a result of these effects, TF is often added to food products as an ingredient, such as dairy products, ice cream, biscuits, etc. [4,5,6]. Its chemical components are often extracted and used as thickeners, gelling agents, etc., in food processing. Hu et al. [7] and Tang et al. [8] isolated the active components in TF and applied them as emulsifiers and thickeners to compound drinks, yogurt and other foods. 

Noodles are staple foods in the Asian diet. Fresh noodles are especially preferred, and increasingly, consumers are paying more attention to the quality and taste of noodles. Therefore, food additives such as hydrocolloids are widely used in noodle products. Hydrocolloids such as sodium alginate, guar gum and xanthan gum are frequently used in noodle products because they can modify and control physical properties. Sodium alginate (SA) is an important polysaccharide extracted from the cell walls of brown seaweed. Adding SA to noodles can contribute to a uniform and compact structure, and improve the rheological properties of dough, and texture properties of noodles [9]. Guar gum (GG) is a water-soluble non-ionic polysaccharide, which can increase the tensile strength and elasticity of noodles, and decrease the cooking loss [10]. Xanthan gum (XG) is an extracellular heteropolysaccharide secreted by the microorganism *xanthomonas campestris*. Adding XG to noodles can increase dough consistency, resulting in fresh noodles with higher hardness and firmness, and reduced cooking loss [11]. Currently, people pay more attention to natural, functionally healthy food. The results showed that the addition of fungus could strengthen noodles’ quality and nutritional characteristics [12,13]. If TF can be added to noodles as a thickening agent, it could improve the quality of the noodles, slow down the digestion of starch in noodles, avoid the rapid rise of blood sugar after meals, help prevent the occurrence of diabetes, and enrich the flavor of noodles.

At present, scholars mainly focus on the functional activity of TFP. There are relatively few studies on the application of TF in noodle products, and whether it can produce similar effects as some hydrocolloids in noodle making. Therefore, the objectives of this work are: (1) to evaluate the feasibility of producing noodles at different levels of TF addition (1%, 2%, 3%, 4%); (2) to investigate the effect of TF, SA, GG and XG on the rheological properties of dough and the starch digestibility of noodles; and (3) to explore the mechanism of how TF and hydrocolloids modify the end-use quality of noodles from the perspectives of protein secondary structure, thermal stability and moisture migration. This study can help improve the application of *Tremella fuciformis* in noodle products, and lays the foundation for the production of natural, functionally healthier noodle products with acceptable qualities. 

## 2. Materials and Methods

### 2.1. Materials and Chemicals

*Tremella fuciformis* (TF) was purchased from Gutian, Fujian (China), consisting of 9.63% protein, 3.03% fat, 10.34% moisture, and 4.83% ash. Wheat flour was purchased from Jinyuan Food Co., Ltd. (Zhengzhou, China), consisting of 11.11% protein, 1.56% fat, 13.69% moisture, and 0.48% ash. Sodium alginate (SA), guar gum (GG), and xanthan gum (XG) were purchased from Xinfeng Laboratory Co., Ltd. (Zhengzhou, China). Amounts of 1,1-diphenyl-2-picrylhydrazyl (DPPH), 2,2′-azino-bis (3-ethylbenzothiazoline-6-sulfonic acid (ABTS), trypsin, urea, and starch glucosidase were purchased from McLean Reagent Co., Ltd. (Shanghai, China).

### 2.2. Dough and Noodles Preparation 

The basic formula for the noodles was 100 g of wheat flour and 37 g of distilled water. On a flour weight basis, different amounts of TF powder (1%, 2%, 3%, 4%), SA (0.2%, 0.4%, 0.6%, 0.8%), GG (0.2%, 0.4%, 0.6%, 0.8%) or XG (0.2%, 0.4%, 0.6%, 0.8%) were added separately. TF, SA, GG and XG solutions were prepared by separate dispersal in water at room temperature, prior to adding to wheat flour. A control noodle dough was also prepared, consisting of 100 g of wheat flour and 37 g of distilled water. Each dough sample was prepared by blending all of its components in a dough mixer (JHMZ-200, Dongfu Jiuheng Instrument Technology Co., Ltd., Beijing, China) at 104 rpm for 6 min. The dough was rested at 25 °C for 15 min. Then the dough billet was repeatedly passed through a multi-functional noodle machine (Dongfu Jiuheng Instrument Technology Co., Ltd., Beijing, China) five times to obtain noodle samples of 3 mm in width and 1 mm in thickness.

### 2.3. Texture Profile Analysis (TPA)

TPA of the different samples were determined using a Texture Analyzer (TA-XT Plus, Stable Micro Systems, London, UK). The fresh noodles were boiled for the optimum cooking time and rinsed with cold water for 20 s, drained, then three noodles were placed on the special stage equipped with the texture analyzer and tested using the HDP/PFS probe. Three parallel experiments were conducted. Parameter settings were: pre-test speed, 1 mm/s; test and post-test speed, 5 mm/s; and strain displacement, 75%.

### 2.4. Cooking Properties

The cooking properties of noodles were determined by previously reported methods [13]. Raw noodles (10 g) were put into 500 mL of boiling distilled water for the optimal cooking time, and the optimal cooking time was defined as the time at which the white core in the central portion of the noodle strand disappeared when the noodle was squeezed between two transparent glass plates. Then, cooked noodles were rinsed in cold water for 30 s, drained, and weighed to obtain the water absorption (%). Meanwhile, the cooking water and rinse water were collected in a volumetric flask and the volume was adjusted to 500 mL with distilled water. This was then transferred to a beaker, where most of the water was evaporated on an infrared heater, and dried to a constant weight at 105 °C, to obtain the cooking loss (%). 

The water absorption of the sample was calculated according to Equation (1):(1)$$\mathrm{Water}\;\mathrm{absorption}\;(\%)=\frac{m_{1}-m_{0}}{m_{0}}\;\times \;100\%$$
where *m*_0_ was the weight of noodles before cooking (g), and *m*_1_ was the weight of noodles after cooking (g).

The cooking loss of the sample was calculated according to Equation (2): (2)$$\mathrm{Cooking}\;\mathrm{loss}\;(\%)=\frac{m_{4}-m_{3}}{m_{2}}\;\times \;100\%$$
where *m*_2_ was the weight of noodles before cooking (g), *m*_3_ was the empty pre-dried beaker weight (g), and *m*_4_ was the residue and beaker weight (g).

### 2.5. Dynamic Rheological Measurements of Dough

The rheological properties of the samples were tested by a dynamic shear rheometer (MARS600, Thermo Fisher Scientific, Waltham, MA, USA) equipped with a 30 mm parallel plate geometry at a gap of 2 mm, and a rheometer plate at 25 °C. The sample was loaded between the plates for 5 min for equilibration, and the extra dough was trimmed. To prevent dough dehydration during the tests, the edge of the dough was coated with a thin layer of silicone oil. Elastic modulus (G′), viscous modulus (G″) and tan δ (G″/G′) were determined at a constant shear strain of 0.1% (within the linear viscoelastic region) with a frequency sweep from 0.1 to 20 Hz.

### 2.6. Protein Secondary Structure 

The secondary structures of the different samples were characterized by using a Fourier transform infrared spectroscopy (FTIR) spectrometer (WQF-510, Beijing Ruili Analytical Instrument Co., Ltd., Beijing, China) according to the method of Ge et al. [14]. Amounts of 2 mg samples and 200 mg of potassium bromide were mixed, ground, and dried, then pressed into translucent pellets using a hydraulic press at 24 MPa for 1 min. For the FTIR spectroscopy measurements, the resulting FTIR spectra at 1600–1700 cm^−1^ (amide I band) were analyzed using PeakFit software (version 4.12, Systat Software Inc., San Jose, CA, USA).

### 2.7. Thermogravimetric Analysis 

The thermal properties of the samples were examined by thermogravimetric analysis (TGA/SDTA851E, Mettler Toledo Corp., Zurich, Switzerland). Freeze-dried noodle samples (10 mg) were placed into a crucible. The samples were heated from 20 °C to 600 °C at a constant rate of 20 °C/min under a nitrogen atmosphere with a flow rate of 20 mL/min. 

### 2.8. Water Distribution 

The water distribution of different samples was acquired using a low-field nuclear magnetic resonance analyzer (LF-NMR, MicroMR-CL-I, Shanghai Niumai Electronic Technology Co., Ltd., Shanghai, China) according to the method provided by Zheng et al. [15]. Approximately 1.0 g noodle samples were weighed, formed into 30 mm strips, wrapped with a special cloth, put in the special tube for NMR, and placed at the center of the permanent magnetic field for testing. Different noodle samples were laid in sealed bags and placed at 25 °C for 0 h, 4 h and 8 h, respectively, then repeated. The test parameters were set as follows: interval time of sampling (TW), 1500 ms; echo time (TE), 0.10 ms; number of echoes (NECH), 3000; and number of scans (NS), 32. The CPMG data were collected and analyzed by T2-fit firm software.

### 2.9. In Vitro Digestion Properties of Cooked Noodles

Refer to the method of Sun et al. [16]. The cooked noodles (1 g) were mixed with 20 mL of acetate buffer (0.1 mol/L, pH = 5.2) in a 50 mL conical flask with glass beads, 5 mL of enzyme solution was added, then the solution was incubated at 37 °C with a stirring rate of 190 rpm. After 20 and 120 min of digestion, 0.5 mL of the hydrolysate was collected and mixed with 20 mL of 70% ethanol solution, centrifuged at 4000 r/min for 5 min, then 0.1 mL of the hydrolysate was collected and mixed with 3 mL GOPOD and incubated at 45 °C for 20 min, whereupon the absorbance was measured at 510 nm. Amounts of 1 mg/mL of standard glucose solution and 0.1 mL of distilled water were weighed and mixed with 3 mL of GOPOD, incubated at 45 °C for 20 min, and the absorbance value was measured at 510 nm. The operation was repeated after 120 min. Based on the absorbance values, we calculated the RS (resistant starch), RDS (rapidly digestible starch) and SDS (slowly digestible starch) contents of the sample and standard glucose, respectively. The RDS, SDS, and RS calculation equations are, respectively, shown in Equations (3)–(5): (3)$$\mathrm{RDS}=\frac{A\times 104\times 0.9}{C_{20}}$$
(4)$$\mathrm{SDS}=\frac{\left(B\;-\;A\right)\times 104\times 0.9}{C_{120}}$$
RS = 100 − RDS − SDS(5)
where A and B are the difference between the average absorbance of the sample and the absorbance of the blank control at 20 min and 120 min, respectively; and *C*_20_ and *C*_120_ are the standard glucose absorbance at 20 min and 120 min.

### 2.10. Microstructure 

The microstructure of the transverse cross-section of different noodles was observed by scanning electron microscope (Quanta FEG250, FEI, Hillsboro, ORE, USA) at 2000× *g* magnification. The fresh noodles were freeze-dried and processed into small regular pieces. Each sample was coated with gold in a sputter coater and photographed for observation. 

### 2.11. Statistics Analysis

The data were presented as the mean ± standard deviation (SD) from the duplicate analysis. Statistical significance was analyzed by one-way ANOVA using SPSS 21.0 software (IBM Inc., Armonk, NY, USA). Values with different letters mean significant difference (*p* < 0.05). Figures were drawn using Origin 2017 software (Origin Lab, Inc., Northampton, MA, USA).

## 3. Results and Analysis

### 3.1. Texture Properties

Texture analysis is an effective method for evaluating the quality of noodles. Previous studies have shown that hardness, chewiness and adhesiveness are quality parameters usually associated with sensory evaluation [17]. The effects of TF, SA, GG, and XG, on the hardness, adhesiveness and chewiness of fresh noodles are shown in Figure 1, where a, b and c represent the hardness, adhesiveness and chewiness of different formula noodles after soaking in hot water for 0, 5 and 10 min, respectively. It can be seen from a, b, c, that when TF was added at 1–3%, SA at 0.2–0.6%, and GG and XG at 0.2–0.4%, respectively, the hardness, adhesiveness and chewiness of the noodles showed a significant (*p* < 0.05) increase, and thereafter a decreasing trend. This result was similar to the trend of previous studies, which found that the addition of hydrocolloids (xanthan gum, guar gum, etc.) increased the hardness, adhesiveness, chewiness and resilience of the noodles [10,11]. The increase in hardness and chewiness of the noodles was beneficial in enhancing the noodles’ texture and strength [13]. This indicates that TF can improve the edible quality of the noodles. After adding TF, the increasing trend of noodle hardness, adhesiveness and chewiness was similar to SA and GG, but slightly lower than XG. It may be that the viscosity of the colloidal solution formed by XG is higher than the other three, and its high adhesion promotes the formation of a good dense gluten network structure, which increases the chewiness, hardness and adhesiveness of the noodles. SA and GG are hydrocolloids with specific viscosity when in contact with water, and the polysaccharides and fiber components in TF may intertwine with the gluten structure, increasing the strength of the network structure and, thus, increasing the chewiness and hardness of the noodles. When SA, GG and XG are added in excessive amounts, the high viscosity will soften the noodles and reduce their quality. TF has a strong water-holding capacity, and water may migrate from the gluten to the TF structure when added at high levels (>3%), detrimentally affecting the gluten network formation in noodles [18], resulting in a decrease in hardness, adhesiveness and chewiness. The tensile properties data, shown in Appendix A, yielded similar trends to the textural properties, indicating that there was no significant (*p* < 0.05) difference in the effects of TF, SA and GG on the tensile and textural properties of noodles. 

### 3.2. Cooking Properties

The water absorption and cooking loss of TF, SA, GG and XG noodles are shown in Figure 2a–d. The water absorption of the noodles increased from 91.13% to 107.26% with the addition of TF, while the water absorption decreased from 91.13%, to 80.69%, 80.96% and 78.18%, after adding SA, GG and XG, respectively. TF contains large amounts of gums and polysaccharides, which have high water absorption and water retention properties, thus, increasing the water absorption of the noodles [2]. SA, GG and XG are hydrocolloids, which form a high viscosity liquid after contact with water. When combined with starch, the hydrocolloid may form a film that hinders the gelatinization and water absorption of starch during cooking, resulting in a decrease in water absorption.

Cooking loss rate is often used as an indicator to evaluate the cooking characteristics of noodles [10]. As can be seen in Figure 2, the cooking loss of the noodles increased from 5.25% to 5.43% after adding TF. This may be due to the water-soluble substances in TF dissolving in the noodle soup during cooking, and more TF may also dilute the gluten and weaken the gluten network, causing the starch in the noodles to dissolve out of the noodle soup, resulting in a higher cooking loss rate. The cooking loss rate of noodles with SA, GG and XG added, showed a decreasing trend. The same effect was reported by Hong et al. [9] who showed that adding hydrocolloids to noodles could reduce the dissolution of starch in the cooking process, to reduce the cooking loss of the noodles. When the amount of SA was 0.6%, and GG and XG were 0.4%, the cooking loss rate reached the minimum and then increased. It may be that the gel network structure formed by SA, GG and XG in contact with water produced an adhesive effect, reduced the starch leaching during noodle cooking, and reduced the turbidity of the noodle soup [19]. However, when added in higher amounts, the higher viscosity may have caused partially-condensed structures in the gluten network, destroying the original dense structure and resulting in a slightly higher cooking loss rate.

### 3.3. Dynamic Rheological Measurements of Dough

Food rheological properties play an important role in maintaining the structure and food stability of food products [3]. Based on the previous results, 3% TF, 0.6% SA, 0.4% GG and 0.4% XG were selected for subsequent correlation experiments. The changes in rheological properties of the mixed dough samples as affected by the addition of 3% TF, 0.6% SA, 0.4% GG and 0.4% XG were determined by the oscillation frequency sweep test. As shown in Figure 3, upon the addition of 3% TF or hydrocolloids to the dough, there was a corresponding increase in both G′ and G″, and G′ was much greater than G″ at all values of angular frequency, indicating that all doughs exhibited an elastic characteristic similar to a solid [2]. There are many polysaccharides in TF, which have a linear chemical structure, and they gradually entangle into a solid-like network structure in aqueous solution [20]. This structure can help the system store energy, so that G′ > G″. With the increasing addition of TF, the more entanglement occurs between chains, and the higher the stored modulus. The tan δ value is the ratio of G″ to G′, which reflects the relative contribution of the viscoelastic sample to the viscosity and elastic components [9]. Tan δ values of all samples were less than 1, indicating that elastic characteristics predominated over viscous behavior throughout the frequency range tested. Meanwhile, tan δ values of dough exhibited a decreased trend with the 3% TF or hydrocolloids addition, which suggested 3% TF and hydrocolloids, respectively, have a greater effect on the improvement of G′ as compared with G″, and the solid, elastic characteristics of dough were enhanced with the addition of 3% TF or hydrocolloids. This trend was similar to the experimental results of Tian et al. [21], who found that the G′ of the blend system increased and the tan δ value decreased with the increase in the *Tremella fuciformis* polysaccharides addition ratio. Furthermore, dough with 0.4% XG had the largest G′ and the smallest tan δ value. The G′, G″ and tan δ value after adding 3% TF exhibited no difference from dough with the addition of 0.6% SA or 0.4% GG. This result might be ascribed to the alteration of dough consistency induced by the greater water-binding capacity of XG [15]. The formation of new hydrogen bonds induced by the addition of TF or hydrocolloids could ultimately increase the strength of the gluten network and the viscoelastic properties of the mixed dough [20,22]. The same effect was reported by Farbo et al. [23], who speculated that electrostatic interactions between hydrocolloids and gluten might be connected with the higher G’ value. Based on the rheological properties of TF in Appendix A Appendix A, it can also be seen that the TF solution mainly presented a pseudoplastic characteristic of non-Newtonian fluid and an elastic characteristic similar to solid and weak gel-like behavior, indicating that the addition of TF was beneficial in improving the viscoelastic properties of dough. Moreover, a food with a pseudo-plastic property has an excellent mouth feel, showing a stable sense of flow during chewing and viscosity restoration after chewing that is conducive to swallowing [2]. Therefore, TF has the potential to be applied in noodle products. This result corresponds to the result of noodle texture properties.

### 3.4. Protein Secondary Structure

The amide I band (1600–1700 cm^−1^) in the FTIR spectrum is the region where the protein secondary structure is determined [24]. It was found that α-helix and β-sheet are beneficial to the formation of ordered secondary structures, while β-turn and random coil are not conducive to the ordering of protein secondary structures [14]. The effect of TF and hydrocolloids on the protein secondary structure is shown in Table 1. Compared with the control sample, adding 3% TF increased the β-sheet and α-helix by 1.12% and 1.39%, respectively, and decreased the random coil and β-turn by 0.96% and 1.54%. Ge et al. [14] obtained similar results. They found that when adding alginate to the dough, the total content of random coil and β-turn declined 0.31-fold and 0.1-fold, respectively, while the total content of β-sheet and α-helix increased by 2.29% and 3.02%, respectively. Additionally, the increasing conformation of α-helix may lead to a more ordered structure. The effects of TF on β-sheet, α-helix and β-turn were not significantly (*p* < 0.05) different from those of SA, GG and XG, while the tendency of TF to reduce random coil was weaker than for the three hydrocolloids. This result indicates that the addition of 3% TF was conducive to the stable and orderly protein secondary structure, and the effect of 3% TF on the secondary structure was similar to that of the three hydrocolloids. This trend was generally in agreement with the rheological measurement, suggesting that an enhanced gluten network with a more pronounced viscoelastic property was linked with the protective behavior of TF or hydrocolloids [11]. The polysaccharide component of TF or the hydrophilic group in the hydrocolloid, hydrogen-bonded with water molecules and some amino acid residues, forming a polysaccharide/hydrocolloid-water-polypeptide chain complex, which affects the interaction between water and amino acid side chains, thus, affecting the periodic spatial arrangement of amino acid residues, leading to changes in the protein secondary structure [25]. Delcour and Gao suggested that the α-helix is a solid elastic structure which is related to the elasticity and stiffness of gluten proteins, while the β-sheet is associated with the viscosity of gluten proteins, and both rely on hydrogen bonds to form a stable structure, which has an impact on the quality of noodle products [24,26]. The addition of TF increased the β-sheet and α-helix content, indicating that the introduction of TF may increase the hardness, elasticity and adhesiveness of the noodles, which was consistent with the texture properties results. In addition, the data on protein intermolecular interactions in Appendix A show that the addition of TF or hydrocolloids increased the hydrogen bond content and reduced the hydrophobic interactions, which corroborates with these data.

### 3.5. Thermogravimetric analysis

The characteristic parameters of the thermogravimetric analysis curves implying the structure of the gluten protein are shown in Table 1. It can be seen that compared with the control sample, the degradation temperature of noodles after adding 3% TF, 0.6% SA, 0.4% GG or 0.4% XG increased significantly from 333.00 °C to 341.00 °C, 343.40 °C, 335.40 °C and 348.70 °C, respectively (*p* < 0.05), and the weight loss decreased significantly from 97.47%, to 93.85%, 90.56%, 95.56% and 85.04%, respectively (*p* < 0.05), which indicates that the gluten network structure was formed more solidly, and the thermal stability was elevated, after adding TF or hydrocolloid. The effect of TF in stabilizing the structure was stronger than of GG. This may be due to the hydrogen bonding between the hydroxyl group in TF and the gluten protein and starch, which strengthens the entangling and cross-linking among the three, making starch more evenly embedded in the gluten network and increasing the stability of the network structure [20,25]. In addition, according to the data of protein secondary structure, adding 3% TF improved the orderliness and stability of the network structure and reduced the disordered structure. It caused the network structure to form more tightly, thus, reducing the weight loss rate.

### 3.6. Water Distribution and Migration

LF-NMR uses the spin relaxation characteristics of hydrogen protons in a magnetic field to study the water morphology, distribution and migration of substances through the change of relaxation time. Appendix A shows the distribution of moisture relaxation time T_2_ of the noodles after 0 h with different additives. The three peaks in the curve represent the three forms of moisture present in the samples. T_21_ represents the deeply bound water, mostly bound to large molecules such as starch or gluten proteins, in the sample. T_22_ represents the weakly bound water, which is present between the starch and gluten protein structures [27]. T_23_ represents free water, which exists outside of starch and protein and can flow freely. Table 2 shows the effect of TF and different hydrocolloids on the relaxation time and proportion of the peak area of the noodles when placed at different times. It can be seen that the addition of 3% TF significantly (*p* < 0.05) reduced T_23_ from 117.59 ms to 77.72 ms, which was greater than the reduction in T_23_ by the three hydrocolloids and had no significant effect on T_21_ and T_22_ (*p* < 0.05). After the noodles were placed for 4 and 8 h, the above trend remained. In general, the shorter the relaxation time, the lower the degree of freedom of water, the tighter the binding to the substrate, and the stronger the water retention capacity of the noodles [27]. It showed that 3% TF could reduce the degree of water freedom in noodles and enhance the water holding capacity of noodles. Table 2 showed that the addition of TF or hydrocolloids increased A_21_ to different degrees, indicating that the addition of TF could cause the water in the noodles to migrate to the deep layer and make the combination of water and gluten network more compact, which was similar to the results reported by Zheng et al. [15]. They found that after adding guar gum, the content of deeply bound water in the starch gel increased and the water mobility decreased. With the increase in noodle storage time, A_21_ decreased and A_23_ increased. The water gradually transferred from the interior of the structure to the surface and finally evaporated into the air. By calculating the change difference (Δ) between A_21_ and A_23_ at different storage times, it was found that when the noodles were left for 4 h, ∣ΔA_21_∣: Control > GG = XG > SA > TF, ∣ΔA_23_∣: Control > TF > GG = SA > XG. When placed for 8 h, ∣ΔA_21_∣: Control > XG > GG > SA > TF, ∣ΔA_23_∣: Control > TF > GG > XG > SA (Appendix A). It is suggested that 3% TF can reduce the mobility of water in noodles, reduce the change rate of each moisture, and have a moisturizing effect on noodles, which is similar to the effect of the hydrocolloids on the moisture distribution of noodles. This may be due to the polysaccharide in TF containing many hydroxyl groups, which have strong water-binding ability and can combine with water molecules. During the formation of the gluten network by gluten protein water absorption, polysaccharide molecules combine with gluten protein with the movement of water, which affects the cross-linking effect between gluten proteins, and then affects the spatial arrangement of the gluten network structure [15,20]. Meanwhile, according to the protein secondary structure and thermogravimetric data, the addition of TF could enhance the stability of the network structure, thus, could fix some water molecules in the structure and reduce the water migration and water loss between gluten and starch.

### 3.7. In Vitro Digestion Properties

In vitro digestion reflects the digestion characteristics according to the starch content changes of different components. Table 3 shows the effect of different additives on the starch fraction of the noodles. After adding 3% TF, the RDS content decreased significantly (*p* < 0.05) by 4.13% compared with the control sample, while the SDS and RS content increased significantly (*p* < 0.05) by 2.44% and 1.68%, respectively. The RDS and SDS contents were not significantly (*p* < 0.05) different from those of the noodles with three hydrocolloids. The increase in RS content was the lowest among the four. RDS is the rapidly digestible starch that can quickly be digested by the small intestine, SDS is the slowly digestible starch that is digested slowly by the small intestine, and RS is the resistant starch that cannot be digested by the small intestine. High levels of rapidly digestible starch in food may accelerate the rate of glucose production in the body, leading to an increase in postprandial blood glucose, which is detrimental to human health [28]. The slowly digestible starch can not only maintain the stability of postprandial blood glucose and maintain the sense of satiety, but also improve the body’s sensitivity to insulin and prevent the occurrence of some diseases [29]. The above results showed that the addition of 3% TF could reduce the starch digestibility of boiled noodles, and the effect was similar to that of three hydrocolloids on the digestive characteristics of the noodles. Jenkins et al. found that non-starch polysaccharides and soluble dietary fiber could reduce the digestibility of starch by wrapping the starch [30]. Therefore, it is speculated that the increase in SDS and RS contents in noodles after adding TF may be related to the components of polysaccharide and dietary fiber in TF. Meanwhile, the rheological properties of the TF solution, shown in Appendix A, also showed that the TF solution had good viscosity and exhibited an elastic characteristic similar to weak gel-like behavior, which may affect the binding of digestive enzymes to the active sites, leading to a decrease in the digestibility of starch and an increase in resistant starch content [8,13]. The gel network formed by introducing appropriate amounts of hydrocolloids interacts with starch and protein, stabilizes the structure of starch granules, decreases the degree of starch granules swelling, and entraps starch granules into a physical barrier to reduce starch digestibility [10]. In addition, Appendix A shows that the antioxidant activity of the noodles increased significantly with the addition of 3% TF (*p* < 0.05), while there was no significant change with the addition of hydrocolloids.

### 3.8. Microstructure

The internal structure of noodles is closely related to the quality of noodles. Figure 4 shows the microstructure observation of the noodles with different additives. Figure 4a shows a sample from the control group. It can be seen that the control sample noodles had some holes in the cross-section, the internal structure was discontinuous with faults, and most of the starch was exposed on the surface of the structure and not tightly wrapped. Figure 4b shows that the addition of 3% TF enhanced the continuity of the internal structure, the fracture disappeared, and most of the starch was wrapped inside the structure, making the structure denser and tighter, indicating that the introduction of TF can cause the moisture distribution to become more uniform and the gluten network structure more orderly and stable. Figure 4c–e shows that the addition of 0.6% SA, 0.4% GG, or 0.4% XG can reduce the size of the holes in the internal structure of the noodles, reduce the starch exposed on the surface of the structure, embed the starch in the gluten network, and strengthen the network structure. Among them, the noodle structure with the addition of 0.4% GG showed slight fracture (marked in red in Figure 4d), probably due to the weaker binding capacity of GG to water molecules than the other three, which had a weaker strengthening effect on the gluten network, resulting in the fracture phenomenon not disappearing completely, and, thus, the improvement of the gluten network structure and the textural properties of the GG noodles were relatively weaker.

### 3.9. Pearson’s Correlation Analysis

Table 4 shows the correlation between the textural properties, protein secondary structure, thermal properties and moisture morphology of the noodles with different additives (control, 3% TF, 0.6% SA, 0.4% GG, and 0.4% XG). It can be seen that the hardness of noodles was significantly and positively correlated with α-helix, degradation temperature and deeply bound water content (*p* < 0.05); and negatively correlated with β-turn, random coil, weight loss rate and weakly bound water content. Chewiness was significantly and positively correlated with α-helix, β-sheet, degradation temperature and deeply bound water content (*p* < 0.05); and negatively correlated with random coil and weight loss rate. Adhesiveness was significantly and positively correlated with β-sheet content (*p* < 0.05). This indicated that the protein secondary structure, thermal properties and moisture morphology of noodles were closely related to the textural properties. The polysaccharides in TF contain a large number of hydroxyl groups, which have strong water binding ability and can bind with water molecules. In the process of gluten protein absorbing water to form a gluten network, TFP molecules will interact with gluten protein and starch by hydrogen-bonding with water migration, which increases the cross-linking between the three and causes the gluten network to more closely wrap with starch, thus, the content of ordered structures such as α-helix and β-sheet increases, which improves the orderliness of the gluten network, and at the same time enhances the thermal stability of the gluten network and reduces the weight loss rate. The enhanced stability of the network structure can also fix some water molecules in the structure, the water can be more tightly bound to other components, and reduce water migration and water loss between the gluten protein and starch. A stable and orderly gluten network is beneficial to improving the hardness and chewiness of the noodles, which, in turn, improves the overall quality of the noodles.

## 4. Conclusions

The addition of TF had a positive effect on the textural properties, protein properties and moisture distribution of the noodles. Compared with the control sample, the addition of 3% TF significantly (*p* < 0.05) increased the hardness, adhesiveness, and chewiness of the noodles, and had better texture properties. The addition of 3% TF enhanced the storage modulus (G′), and loss modulus (G″) of dough, which is beneficial to improving the viscoelasticity of dough. Adding TF significantly (*p* < 0.05) increased the degradation temperature, the content of α-helix, β-sheet and deeply bound water in the noodles, and reduced the weight loss, content of random coil and relaxation time, which were conducive to improving the orderliness and stability of the gluten network structure and reducing moisture mobility. Additionally, TF reduced the change rate of the three types of water in the noodles during storage for 0–8 h, further suggesting that TF could restrict water migration in noodles. Furthermore, 3% TF increased the contents of RDS and RS, and reduced the starch digestibility of boiled noodles. Compared with the hydrocolloids, 3% TF was slightly weaker than 0.4% XG in the texture properties of noodles, and had no significant difference from 0.6% SA and 0.4% GG (*p* < 0.05). The effects of 3% TF on structural properties, moisture distribution and digestive characteristics of the noodles were not significantly different from those of the three hydrocolloids (*p* < 0.05). Our results indicated that TF had similar functions to the three hydrocolloids in noodles and could be used as a potential natural hydrocolloid in the processing of noodles.

## Figures and Tables

**Figure 1 foods-11-02617-f001:**
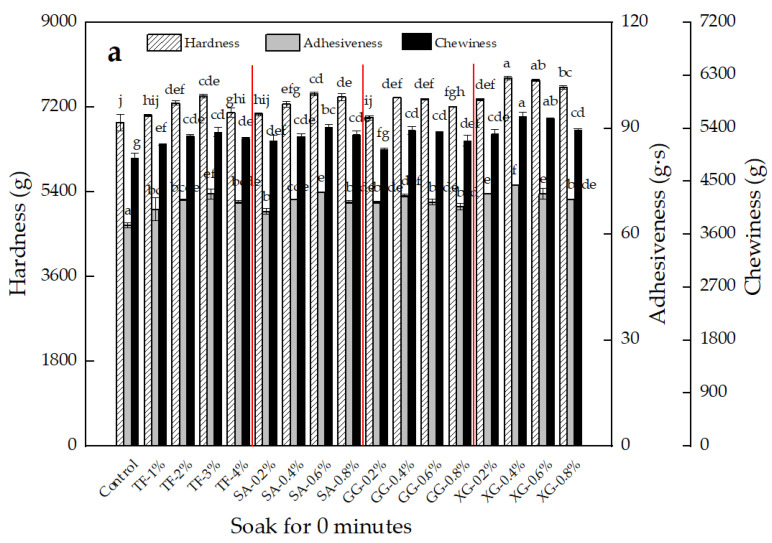
Effect of TF, SA, GG and XG on the textural properties of fresh noodles ((**a**): noodles soaked for 0 min after cooking, (**b**): noodles soaked for 5 min after cooking, (**c**): noodles soaked for 10 min after cooking). Note: All experiments were carried out in triplicate according to independent replicate experiments, and the results were reported as mean value ± standard deviation. Values with the same superscript letters in in different figures are not significantly different at *p* < 0.05.

**Figure 2 foods-11-02617-f002:**
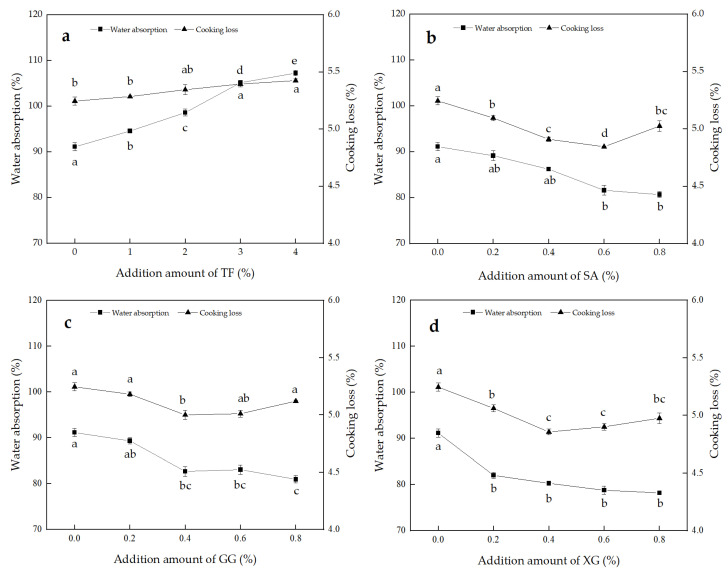
Effect of TF (**a**), SA (**b**), GG (**c**), and XG (**d**), on water absorption and cooking loss of noodles. Note: All experiments were carried out in triplicate according to independent replicate experiments, and the results were reported as mean value ± standard deviation. Values with the same superscript letters in in different figures are not significantly different at *p* < 0.05.

**Figure 3 foods-11-02617-f003:**
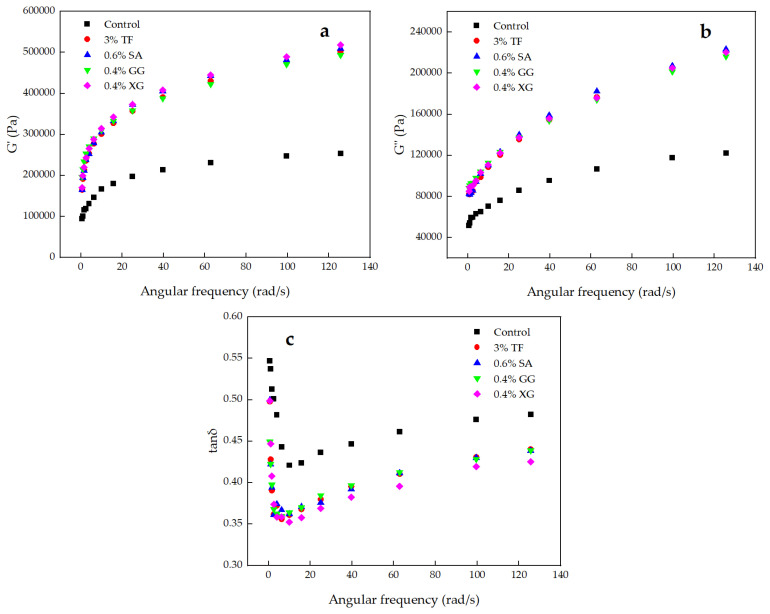
Effect of TF, SA, GG and XG on the (**a**) storage modulus (G′), (**b**) loss modulus (G″), and (**c**) loss tangent (tan δ) of dough.

**Figure 4 foods-11-02617-f004:**
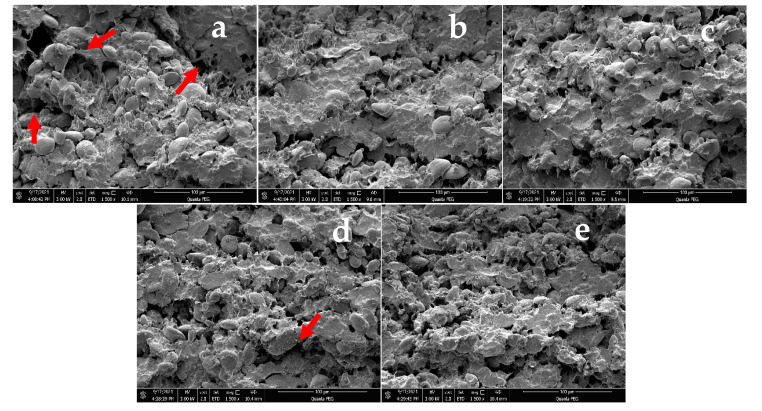
SEM images of noodles with different additives: (**a**) control, (**b**) 3% TF, (**c**) 0.6% SA, (**d**) 0.4% GG, (**e**) 0.4% XG. Note: The red arrows in the figure indicate that there are discontinuities in the structure and the starch is not tightly wrapped.

**Table 1 foods-11-02617-t001:** Effect of TF, SA, GG and XG on the secondary structure of proteins and TGA characteristic parameters.

Sample	Secondary Structures				TGA	
	β-Sheet/%	Random Coil/%	α-Helix/%	β-Turn/%	Degradation Temperature/°C	Weight Loss/%
Control	33.95 ± 0.99 ^c^	18.23 ± 0.09 ^a^	17.16 ± 0.78 ^b^	30.66 ± 0.30 ^a^	333.00 ± 0.40 ^e^	97.47 ± 0.21 ^a^
3% TF	35.07 ± 0.13 ^ab^	17.27 ± 0.04 ^b^	18.55 ± 0.03 ^a^	29.12 ± 0.07 ^b^	341.00 ± 0.10 ^c^	93.85 ± 0.17 ^c^
0.6% SA	34.68 ± 0.20 ^bc^	16.67 ± 0.10 ^c^	18.50 ± 0.61 ^a^	30.15 ± 0.71 ^ab^	343.40 ± 0.30 ^b^	90.56 ± 0.20 ^d^
0.4% GG	35.41 ± 0.06 ^a^	16.66 ± 0.01 ^c^	17.78 ± 0.25 ^ab^	30.15 ± 0.21 ^ab^	335.40 ± 0.30 ^d^	95.56 ± 0.14 ^b^
0.4% XG	35.60 ± 0.26 ^a^	16.50 ± 0.34 ^c^	18.53 ± 0.29 ^a^	29.38 ± 0.21 ^ab^	348.70 ± 0.20 ^a^	85.04 ± 0.19 ^e^

Note: All experiments were carried out in triplicate according to independent replicate experiments, and the results were reported as mean value ± standard deviation. Values with the same superscript letters in a column are not significantly different at *p* < 0.05.

**Table 2 foods-11-02617-t002:** The changes of relaxation time T_2_ and peak area ratio of TF, SA, GG and XG noodles under different storage times.

Treatment	Sample	Relaxation Time (T_2_)			Proportion of Peak Area			∣Δ∣ = A4 h/A8 h − A0 h	
		T_21_/ms	T_22_/ms	T_23_/ms	A_21_/%	A_22_/%	A_23_/%	ΔA_21_/%	ΔA_23_/%
0 h	Control	0.12 ± 0.01 ^a^	7.84 ± 0.00 ^a^	117.59 ± 0.00 ^a^	16.54 ± 0.15 ^c^	82.70 ± 0.15 ^a^	0.77 ± 0.00 ^c^	/	/
	3% TF	0.12 ± 0.01 ^a^	6.37 ± 0.00 ^c^	77.72 ± 5.39 ^d^	17.34 ± 0.03 ^a^	81.69 ± 0.13 ^c^	0.85 ± 0.01 ^b^	/	/
	0.6% SA	0.13 ± 0.00 ^a^	7.32 ± 0.00 ^b^	113.64 ± 3.94 ^b^	17.43 ± 0.08 ^a^	81.43 ± 0.40 ^c^	0.82 ± 0.01 ^c^	/	/
	0.4% GG	0.12 ± 0.01 ^a^	7.84 ± 0.00 ^a^	102.34 ± 0.00 ^c^	16.88 ± 0.10 ^b^	82.18 ± 0.07 ^ab^	0.95 ± 0.04 ^a^	/	/
	0.4% XG	0.11 ± 0.00 ^a^	7.58 ± 0.26 ^ab^	113.64 ± 3.94 ^b^	17.14 ± 0.05 ^a^	82.05 ± 0.04 ^ab^	0.81 ± 0.01 ^bc^	/	/
4 h	Control	0.11 ± 0.00 ^a^	6.83 ± 0.00 ^a^	98.91 ± 3.43 ^b^	16.18 ± 0.13 ^c^	82.61 ± 0.11 ^a^	0.98 ± 0.01 ^ab^	0.36	0.21
	3% TF	0.11 ± 0.00 ^a^	5.54 ± 0.00 ^c^	69.91 ± 2.43 ^d^	17.24 ± 0.04 ^a^	81.70 ± 0.06 ^c^	0.99 ± 0.02 ^ab^	0.11	0.15
	0.6% SA	0.10 ± 0.00 ^a^	6.83 ± 0.00 ^a^	102.34 ± 0.00 ^a^	17.25 ± 0.02 ^a^	81.38 ± 0.03 ^d^	0.96 ± 0.06 ^b^	0.18	0.14
	0.4% GG	0.12 ± 0.01 ^a^	6.37 ± 0.00 ^ab^	89.07 ± 0.00 ^c^	16.66 ± 0.01 ^b^	82.11 ± 0.10 ^b^	1.09 ± 0.03 ^a^	0.23	0.14
	0.4% XG	0.10 ± 0.00 ^a^	6.83 ± 0.00 ^a^	98.91 ± 4.43 ^b^	16.91 ± 0.10 ^b^	82.00 ± 0.12 ^b^	0.93 ± 0.03 ^b^	0.23	0.12
8 h	Control	0.11 ± 0.00 ^a^	6.83 ± 0.00 ^a^	95.48 ± 0.00 ^a^	15.83 ± 0.02 ^e^	82.95 ± 0.04 ^a^	1.18 ± 0.02 ^a^	0.71	0.41
	3% TF	0.11 ± 0.00 ^a^	5.54 ± 0.00 ^c^	65.21 ± 2.26 ^d^	17.12 ± 0.03 ^a^	82.03 ± 0.26 ^b^	1.11 ± 0.01 ^a^	0.23	026
	0.6% SA	0.10 ± 0.00 ^a^	6.37 ± 0.00 ^ab^	95.48 ± 0.00 ^a^	17.14 ± 0.02 ^a^	81.84 ± 0.11 ^b^	1.00 ± 0.04 ^b^	0.29	0.18
	0.4% GG	0.12 ± 0.00 ^a^	6.37 ± 0.00 ^ab^	89.07 ± 0.00 ^c^	16.49 ± 0.02 ^c^	82.32 ± 0.01 ^b^	1.19 ± 0.01 ^a^	0.39	0.24
	0.4% XG	0.10 ± 0.00 ^a^	6.83 ± 0.00 ^a^	92.28 ± 3.20 ^b^	16.81 ± 0.01 ^b^	82.18 ± 0.01 ^b^	1.02 ± 0.01 ^b^	0.33	0.21

Note: All experiments were carried out in triplicate according to independent replicate experiments, and the results were reported as mean value ± standard deviation. Values with the same superscript letters in a column are not significantly different at *p* < 0.05.

**Table 3 foods-11-02617-t003:** Effect of TF, SA, GG and XG on in vitro digestibility of noodles.

Sample	RDS/%	SDS/%	RS/%
Control	56.87 ± 0.13 ^a^	27.13 ± 0.12 ^c^	16.01 ± 0.01 ^e^
3% TF	52.74 ± 0.12 ^bc^	29.57 ± 0.13 ^ab^	17.69 ± 0.01 ^d^
0.6% SA	52.47 ± 0.12 ^c^	29.37 ± 0.13 ^ab^	18.17 ± 0.01 ^a^
0.4% GG	53.02 ± 0.12 ^b^	29.16 ± 0.13 ^b^	17.83 ± 0.02 ^c^
0.4% XG	52.33 ± 0.12 ^c^	29.77 ± 0.13 ^a^	17.91 ± 0.02 ^b^

Note: In the table, RDS is rapidly digestible starch, SDS is slowly digestible starch, and RS is resistant starch. All experiments were carried out in triplicate according to independent replicate experiments, and the results were reported as mean value ± standard deviation. Values with the same superscript letters in a column are not significantly different at *p* < 0.05.

**Table 4 foods-11-02617-t004:** Pearson’s correlation analysis between texture, structural characteristics and moisture morphology of TF, SA, GG and XG noodles.

	Hardness	Adhesiveness	Chewiness
α-helix	0.887 *	0.875	0.881 *
β-sheet	0.866	0.899 *	0.886 *
β-turn	−0.888 *	−0.831	−0.769
Random coil	−0.901 *	−0.836	−0.950 *
Weight loss	−0.881 *	−0.761	−0.887 *
Degradation temperature	0.896 *	0.812	0.892 *
A_21_	0.912 *	0.803	0.881 *
A_22_	−0.899 *	−0.856	−0.863

Note: In the table, * Correlation is significant at the 0.05 level, *p* < 0.05.

## Data Availability

All data included in this study are available upon request by contact with the corresponding author.

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
