# Peer review of "Effect of Tremella fuciformis and Different Hydrocolloids on the Quality Characteristics of Wheat Noodles"

_foods, 2022, doi:10.3390/foods11172617_

Round 1

Reviewer 1 Report

I find the subject of the manuscript interesting and quite original. The authors conducted unconventional measurements of the samples investigated (dough, cooked noodle), such as determination of the protein secondary structure, water distribution and migration, digestion properties, in order to complement the standard measurements.

Pay attention to the font size in the text (Latin names and some of references are written in larger font).

Arrange the references alphabetically.

Section 2.2: How was the percentage amount of TF or hydrocolloid calculated – in reference to all dough components or in reference to amount of flour?

L87: Replace “37%” with “37 g”.

L96: For determination of what?

L110: Replace “quality” with “quantity”.

Equation 1: Reconsider the description of symbols or correctness of the formula (now there is m < m1).

L113-114: Incorrect description of the symbols used in Eq. 2.

L180: Replace “sensory” with “sensory ones” (if it is about it).

Figure 1. Try to enlarge the figures.

Figure 2. Use the same range of values on the ordinate in each of Figure.

L264: There should be comma instead of dot.

L267: Replace “exhibit” with “exhibited”.

Figure 3. Did really values of modulus G’ and G” were up to 500000 Pa and 220000 Pa, respectively?

L405: What do you mean by “wrapping the food”?

L409: What is “weak gelation”?

L414: What is “swelling of starch gelatinization”?

L484-485: Incomprehensible sentence.

Reviewer 2 Report

In this manuscript, Tremella fuciformis (TF) and conventional hydrocolloids such as sodium alginate (SA), guar gum (GG), and xanthan gum (XG) were added to wheat noodles and their influences on the physicochemical and digestibility of the product were evaluated. The results revealed that the incorporation of 3% TF has positive effects on the quality of noodles. The manuscript is well written and the topic is interesting, but the language should be edited by a professional English editor.

Other comments:  

Line 13: Compared

Lines 41-42: Correct the sentence.

Line 68: modify

Lines 84-87: Paraphrase

Line 110: Quality or weight?

Line 113: What are the m3 and m?

Line 119: for equilibration

Figure 1: The graphs are not clear. Please write the data in a table.

Figure 3: Write “control” instead of “blank”. Please apply to the whole manuscript.

Line 329: Write the names of hydrocolloids.

Figure 4: The letters (a, b, c, d, and e) are not written on the images.
